

# Identification and validation of calcium extrusion-related genes prognostic signature in colon adenocarcinoma

Mingpeng Jin[1], Chun Yin[2], Jie Yang[3], Xiaoning Yang[4], Jing Wang[5], Jianjun Zhu[6] and Jian Yuan[1]

[1] Department of Biochemistry and Molecular Biology, Tongji University School of Medicine, Shanghai, China
[2] Department of Cardiology, Xinqiao Hospital, Third Military Medical University (Army Medical University), Chongqing, China
[3] Department of Thoracic Surgery, The First Affiliated Hospital of Nanchang University, Nanchang, China
[4] Institute of Materia Medica, Chinese Academy of Medical Science & Peking Union Medical College, Beijing, China
[5] Department of Cardiology, the 902nd Hospital of PLA Joint Service Support Force, Bengbu, China
[6] Department of Medical Cellular Biology and Genetics, Shanxi Medical University, Taiyuan, China

Corresponding authors
Jianjun Zhu, scdx007@163.com
Jian Yuan, yuanjian229@hotmail.com

## ABSTRACT

**Background**. Disruptions in calcium homeostasis are associated with a wide range of diseases, and play a pivotal role in the development of cancer. However, the construction of prognostic models using calcium extrusion-related genes in colon adenocarcinoma (COAD) has not been well studied. We aimed to identify whether calcium extrusion-related genes serve as a potential prognostic biomarker in the COAD progression.

**Methods**. We constructed a prognostic model based on the expression of calcium extrusion-related genes (SLC8A1, SLC8A2, SLC8A3, SLC8B1, SLC24A2, SLC24A3 and SLC24A4) in COAD. Subsequently, we evaluated the associations between the risk score calculated by calcium extrusion-related genes and mutation signature, immune cell infiltration, and immune checkpoint molecules. Then we calculated the immune score, stromal score, tumor purity and estimate score using the Estimation of STromal and Immune cells in MAlignant Tumor tissues using Expression data (ESTIMATE) algorithm. The response to immunotherapy was assessed using tumor immune dysfunction and exclusion (TIDE). Finally, colorectal cancer cells migration, growth and colony formation assays were performed in RKO cells with the overexpression or knockdown SLC8A3, SLC24A2, SLC24A3, or SLC24A4.

**Results**. We found that patients with high risk score of calcium extrusion-related genes tend to have a poorer prognosis than those in the low-risk group. Additionally, patients in high-risk group had higher rates of KRAS mutations and lower MUC16 mutations, implying a strong correlation between KRAS and MUC16 mutations and calcium homeostasis in COAD. Moreover, the high-risk group showed a higher infiltration of regulatory T cells (Tregs) in the tumor microenvironment. Finally, our study identified two previously unreported model genes (SLC8A3 and SLC24A4) that contribute to the growth and migration of colorectal cancer RKO cells.

**Conclusions**. Altogether, we developed a prognostic risk model for predicting the prognosis of COAD patients based on the expression profiles of calcium extrusion-related genes, Furthermore, we validated two previously unreported tumor suppressor genes (SLC8A3 and SLC24A4) involved in colorectal cancer progression.

## INTRODUCTION

Colon cancer is one of the most common cancers and has the second highest mortality rate of all cancers (*Siegel et al., 2022*). There has been a twofold increase in colorectal cancer cases and deaths worldwide in the last thirty years. In addition, there has been a gradual rise in the prevalence of colon adenocarcinoma (COAD) in China (*Xia et al., 2022*). Several risk factors have been identified in the initiation and progression of COAD, including a sedentary lifestyle, the intake of processed food and alcohol, and smoking (*Dariya et al., 2020*; *Ahmad et al., 2021*). However, the pathogenesis of COAD has not been well elucidated and more effective therapeutic targets for COAD should be pursued.

Calcium ions ($Ca^{2+}$) serve as an important secondary messenger that regulates a variety of cellular functions, including cellular contraction, secretion, metabolism, gene expression, survival, and cell death (*Berridge, Lipp & Bootman, 2000*; *Rimessi et al., 2020*; *Zheng et al., 2023*; *Eisner et al., 2023*). The dysregulation of $Ca^{2+}$ signaling contributes to carcinogenesis and promotes tumor development (*Bettaieb et al., 2021*; *So et al., 2019*; *Silvestri et al., 2023*; *Yang et al., 2019*; *Monteith, Prevarskaya & Roberts-Thomson, 2017*; *Iamshanova, Pla & Prevarskaya, 2017*; *Marchi & Pinton, 2016*). NCXs, NCKXs, and NCLX are crucial ion transport proteins that play a significant role in $Ca^{2+}$ signaling and maintaining $Ca^{2+}$ homeostasis in physiological and pathophysiological conditions (*Khananshvili, 2013*; *Rodrigues, Estevez & Tersariol, 2019*; *Schnetkamp, 2013*; *Luongo et al., 2017*).

The SLC8 gene family that encodes $Na^+/Ca^{2+}$ exchangers (NCXs) belong to the CaCA ($Ca^{2+}$/Cation antiporter) superfamily. Three mammalian proteins (SLC8A1, SLC8A2, and SLC8A3) mediate $Ca^{2+}$-fluxes across the cell-membrane and thus significantly contribute to regulation of $Ca^{2+}$-dependent events (*Khananshvili, 2013*; *Lytton, 2007*; *Rose, Ziemens & Verkhratsky, 2020*). Meanwhile, NCKX6 is known to be the mitochondrial $Na^+/Ca^{2+}$ exchanger; it is also referred to as NCLX and encoded by SLC8B1. NCLX plays a critical role in the regulation release of $Ca^{2+}$ from mitochondria, which is a process heavily controlled by mitochondrial pH, membrane potential, $Ca^{2+}$ levels, and kinase-mediated phosphorylation (*Khananshvili, 2013*; *Luongo et al., 2017*). As a consequence, NCLX function is essential for mitochondrial health, and its dysregulation or loss of function are associated with serious disease and death (*Garbincius & Elrod, 2022*; *Meng et al., 2023*). In humans, SLC8A1 is encoded by a cluster of genes located on chromosome 2p22.1, SLC8A2 is encoded by the gene located on chromosome 19q13.32, SLC8A3 is encoded by the gene located on chromosome 14q24.2, and SLC8B1 is encoded by the gene located on chromosome 12q24.13.

The NCKXs ($Na^+/Ca^{2+}$–$K^+$ exchangers) (SLC24) branch of $Na^+/Ca^{2+}$ exchangers transport $K^+$ and $Ca^{2+}$ in exchange for $Na^+$. There are five members of the NCKX family (NCKX1-5, products of the SLC24A1-5 genes) that exist as a clade in the CaCA superfamily. All NCKX transporters are thought to couple the transport of four $Na^+$ ion in exchange

for one Ca$^{2+}$ ion and one K$^+$ ion (*Vinberg, Chen & Kefalov, 2018*; *Al-Khannaq & Lytton, 2022*; *Hassan & Lytton, 2020*). SLC24A1 is encoded by the gene located on chromosome 8q22; SLC24A2 is encoded by the gene located on chromosome 9p22.1-p21.3; SLC24A3 is encoded by the gene located on chromosome 20p11.23; SLC24A4 is encoded by the gene located on chromosome 14q32.12; and SLC24A5 is encoded by the gene located on chromosome 15q21.1.

Altering the expressions or activities of calcium extrusion related genes have been linked to tumor development (*Ding et al., 2020*; *Muñoz et al., 2015*; *Qu et al., 2017*; *Pathak et al., 2020*). For instance, the downregulation of SLC8A1 modulates calcium concentration to induce apoptosis and proliferation in penile carcinoma (*Muñoz et al., 2015*). The SLC8A2 gene acts as a tumor suppressor and inhibits the invasion, growth, and angiogenesis of glioblastomas (*Qu et al., 2017*). SLC8B1 downregulation causes mtCa$^{2+}$ overload, which increased mitochondrial reactive oxygen species accumulation with consequent HIF1 $\alpha$ signaling activation, leading to the metastasis of SLC8B1-null tumor cells (*Pathak et al., 2020*). Given the vital functions of individual calcium extrusion-related genes in the initiation and progression of tumor, constructing a prognostic model based on calcium extrusion-related genes might be an effective strategy to precisely predict the prognosis in COAD. Herein, a risk model based on calcium extrusion-related genes was developed to predict the prognosis and immunotherapy outcomes for patients with COAD. Additionally, we utilized this risk model to conduct a comprehensive analysis, including the associations between the risk score and mutation profile, tumor mutation burden, immune infiltration, the level of immune checkpoint molecules and the response to immunotherapy. Finally, functional studies showed that upregulation of SLC8AA3 and SLC24A4 inhibited the survival and migration of CRC cells *in vitro*. Taken together, our findings highlight the functional roles of calcium extrusion-related signatures and uncover latent prognostic biomarkers for COAD.

## MATERIAL AND METHODS

### Colon adenocarcinoma data from the TCGA and GEO databases

The transcriptomic data was obtained from the TCGA (https://gdc.cancer.gov/). A total of 524 samples were downloaded from TCGA-COAD, and the samples with incomplete clinical information, including age, gender, T stage, N stage, M stage, and TNM stage, and the overall survival time less than 30 days were excluded from this study. We included 471 samples, comprising 430 tumor samples and 41 healthy samples in the present study.The transcriptional expressions of nine calcium extrusion-related genes were examined in both tumor tissues and normal tissues (*Tomczak, Czerwińska & Wiznerowicz, 2015*). Table S1 displayed comprehensive details of nine genes. GSE17536, GSE29623, and GSE39582 datasets were used as validation cohorts in the present study (https://www.ncbi.nlm.nih.gov/geo/) (*Zheng et al., 2021*; *Liu et al., 2021*). A total of 585 samples were downloaded from GSE39582, and 19 non-tumor samples were excluded in this study. Finally, 566 tumor samples were analyzed in the present study. 177 and 65 tumor samples were analyzed in GSE17536 and GSE29623, respectively. The details were shown in Table S2.

## Clinical-pathological analysis

The clinical-pathological features of COAD patients were obtained from the TCGA database (https://gdc.cancer.gov/). The data is available to the public online and its usage does not violate the rights of individuals or institutions. After dividing the numeric values according to the median, a comparison was made between the low-risk and high-risk groups. A Pearson's chi-square ($\chi 2$) test was used to assess the association between these categorical variables. Additionally, the correlation between the mRNA expression of calcium extrusion-related genes and the pathological stage of COAD patients was examined using Gene Expression Profiling Interactive Analysis (GEPIA, http://gepia.cancer-pku.cn/) (*Cai et al., 2021*).

## Survival analysis in colon adenocarcinoma patients

The relationship between nine calcium extrusion-related genes expressions and the overall survival (OS) and disease-free survival (DFS) of COAD patients was investigated through the Kaplan–Meier Plotter tool (http://kmplot.com/analysis/). Survival analyses were conducted based on the median distribution, with a cutoff of 50% for high expression and 50% for low expression. The hazard ratio value and $p$ value for each gene were uploaded to bioinformatic online software to visualize the prognosis (https://www.bioinformatics.com.cn/).

## The prognosis value of the risk signature

A risk model was built using R software based on the nine calcium extrusion-related genes and the TCGA database was utilized as the training set. The GEO datasets (including GSE39582, GSE17536, and GSE29623) were employed as validation datasets. Patients were categorized into high-risk and low-risk groups based on the median value of their risk score. The disparities in patient survival between these subgroups were assessed using the R packages "survival" and "survminer", and the receiver operating characteristic (ROC) curve was generated using the R package "survival-ROC" to calculate the area under curve (AUC) value. The transcriptional expressions of calcium extrusion-related genes in the high-risk and low-risk groups were visualized using GraphPad Prism 8.0 software. The heatmap was plotted by online tool.

## Analysis of gene mutations, interaction networks, and functional enrichment

The mutation data of patients with COAD were acquired from the cBioPortal (https://www.cbioportal.org/). The tumor mutation burden (TMB) for each sample was determined using the R packages ("maftools"). TCGA-COAD samples were divided into microsatellite stability (MSS), low-frequency microsatellite instability (MSI-L), and high-frequency MSI(MSI-H) using the R packages ("TCGA biolinks"). Protein-protein interaction network analysis was performed using the STRING database (https://cn.string-db.org/), while gene-gene interaction network analysis was carried out using Gene-MANIA (http://genemania.org/). We conducted the functional enrichment of SLC8B1 and its co-expression genes using R packages ("clusterProfiler", "ggplot2", and "enrichplot").

## Immune cell infiltration and responsiveness to immunotherapy

The immune checkpoint genes used in this study were downloaded from a previous study (*Sun et al., 2020*). The relationship between the expressions of immune checkpoint genes and the mRNA expressions of seven calcium extrusion-related genes in COAD was examined. The estimation of immune cell composition was performed using CIBERSORT R package. The list of immune cell signatures was obtained from TISIDB (http://cis.hku.hk/TISIDB/download.php). Sanger box software was used to explore the correlation between the risk score and the expressions of immune cell signature genes (http://www.sangerbox.com/home.html). Additionally, the ESTIMATE algorithm, available in the R package "estimate", was utilized to calculate the estimated score, immune score, stromal score and tumor purity. The anti-PD1 and anti-CTLA4 immunotherapy response in patients with COAD was determined using the TIDE algorithm (http://tide.dfci.harvard.edu/login/) (*Jiang et al., 2018*). PRJEB25780 was used as an immunotherapy validation cohort and was obtained from the European Nucleotide Archive (ENA, https://www.ebi.ac.uk/). The samples were divided into low-risk and high-risk groups based on the median risk score as previously described. For each of the 198 FDA-approved drugs, the IC50 was calculated by through R package ("pRRophetic"). The Genomics of Drug Sensitivity in Cancer database (GDSC, https://www.cancerrxgene.org/) provided information on drugs.

## Cell culture

We purchased human colorectal cancer cell (HCT-116, SW620, SW480, HT-29 and RKO) from the American Type Culture Collection (ATCC, Manassas, VA, USA). SW620 and SW480 were cultured in L-15 supplemented with 10% fetal bovine serum (FBS). HCT-116 and RKO was cultured in DMEM supplemented with 10% FBS. HT-29 was cultured in RPMI-1640 supplemented with 10% FBS.

## Cell transfection, virus packaging and stable cells lines

Cells were transfected with siRNAs using Lipofectamine® 2000 reagent provided by Thermo Fisher Scientific following the manufacturer's instructions. The cells were seeded 50–60% confluent and transfected with 50 nM siRNAs. NC-siRNA was used as the control. The control was NC-siRNA. Shanghai Generay Biotech (China) synthesized all the siRNAs and following is a list of siRNA sequences. SLC8A3 siRNA: sense, 5′-GAAGAGUCCUAUGAGUUCAAGACUA-3′, SLC8A3 siRNA: antisense, 5′-UAGUCUUGAACUCAUAGGACUCUUC-3′; SLC24A2 siRNA: sense, 5′-CAACGUUGGCAUAGGCACAAUUGUA-3′, SLC24A2 siRNA: antisense, 5′-UACAAUUGUGCCUAUGCCAACGUUG-3′; SLC24A3 siRNA: sense, 5′-GGUGCCUGCUGAGGGAUUCUAUUUA-3′, SLC24A3 siRNA: antisense, 5′-UAAAUAGAAUCCCUCAGCAGGCACC-3′; SLC24A4 siRNA: sense, 5′-GAGGCUGGUAAUGAUUUCUAUGACG-3′, SLC24A4 siRNA: antisense, 5′-CGUCAUAGAAAUCAUUACCAGCCUC-3′.

Cells were co-transfected with lentiviral expression vectors and packaging plasmids (psPAX2 and pMD2.G) using Lipofectamine 2000 (Thermo Fisher Scientific, Waltham,

MA, USA) following the manufacturer's protocols. Lentiviral supernatants were collected 48 h after transfection and filtered through a 0.45-micron filter. After transducing the lentiviruses into CRC cells, lentiviruses were added to medium containing polybrene (8 μg/ml). Stable cells were selected using puromycin for 1 week and detected with immunoblotting.

## Total RNA extraction and quantitative real-time PCR

Total RNA extraction was extracted using Trizol reagent (Invitrogen, Waltham, MA, USA) and reverse transcribed with a reverse transcription kit (Takara, Shiga, Japan). Subsequently, qRT-PCR was performed with SYBR Green (Bio-Rad, Hercules, CA, USA) on an ABI PRISM 7900HT detection system (Applied Biosystems, Waltham, MA, USA). The primer sequences used in this study were produced by Generay Biotech (Shanghai, China). Relative mRNA expression levels were quantified by the $2^{-\Delta\Delta Ct}$ method. Experiments were performed in triplicate and GAPDH was used as an internal control. The primer sequences were as follows: SLC8A3 forward sequence 5′–3′: GAGGCCAAGAGGATAGCAGAG and reverse sequence 5′–3′: TGCTGCACTGACGGTGATG; SLC24A2 forward sequence 5′–3′: CCAAGGAGACTACCCGAAAGA and reverse sequence 5′–3′: CAGACAATGGC-TAAGGCTATGAA; SLC24A3 forward sequence 5′–3′: CCCGTGCTGTCCCTTGTATT and reverse sequence 5′–3′: CCCCATCCGTTAATGCTGGT; SLC24A4 forward sequence 5′–3′: TGTGCCGAGACTCCGTGTA and reverse sequence 5′–3′: TGTTGACCGGGT-TACCGTTTG; GAPDH forward sequence 5′–3′: GTCTCCTCTGACTTCAACAGCG and reverse sequence 5′–3′:ACCACCCTGTTGCTGTAGCCA.

## Plasmids, reagents, antibodies

SLC8A3, SLC24A2, SLC24A3 and SLC24A4 were subcloned into PLVX. A DNA sequencing test was performed on all vectors to confirm their validity. Polybrene and puromycin were purchased from Sigma. This research used the following antibodies: anti-GAPDH (60004-1-lg, Proteintech, San Diego, CA, USA), anti-SLC8A3 (ANX-013, Alomone, Jerusalem, Israel), anti-SLC24A2 (ab192419, Abcam), anti-SLC24A3 (ab129776, Abcam, Cambridge, UK), anti- SLC24A4 (118992-1-AP, Proteintech, San Diego, CA, USA). All antibodies were obtained from commercial suppliers.

## Colony formation assay

In each well, $1 \times 10^3$ cells were placed in 6-well plates and left to incubate for 9 days in a culture medium containing 10% FBS. Following staining with 1% crystal violet, colonies per well were counted.

## Cell counting kit-8 assay

We performed a cell counting kit-8 (CCK8) assay using a CCK8 kit according to the manufacturer's instructions (Beyotime). A total of 5,000 RKO cells were evenly distributed in a 96-well plate with a total volume of 100 μL. After incubating for 24 h, 48 h, 72 h or 96 h, each well was supplemented with 10 μL of CCK8 reagent and incubated at 37 °C for 1 h. The plates were analyzed using a 450 nm wavelength.

## Scratch wound healing assay

Cells were seeded in six-well plates and allowed to reach near-complete confluence within a 24-hour period. Subsequently, a wound was created in each well using a 10 µL pipette tip. Following this, the cells were subjected to multiple washes with PBS and subsequently incubated in serum-free culturing medium. The process of wound closure was observed at both 0 and 48 h using a Leica light microscope. The extent of cell migration post-wounding was assessed at 0 and 48 h.

## Statistical analyses

Statistical analyses were conducted using GraphPad Prism 8.0 software. Multiple comparisons were performed utilizing the ANOVA test, while the assessment of the correlation between two variables was carried out using Spearman's rank correlation coefficient. $^*$ $P < 0.05$, $^{**}$ $P < 0.01$, $^{***}$ $P < 0.001$.

# RESULTS

### Bioinformatic analyses revealed mRNA differential expression patterns of calcium extrusion-related genes in COAD

This study process is illustrated in Fig. 1. Firstly, using the RNA-Seq data from TCGA, we found the mRNA expression level of SLC8B1, SLC8A1, SLC8A2, SLC8A3, SLC24A3, and SLC24A4 was significantly decreased, while SLC24A2 was significantly increased in COAD compared with that in normal colon tissues (Fig. 2). Moreover, the similar results of SLC8B1, SLC8A1, SLC8A3, SLC24A4 and SLC24A2 were observed in GSE39582 (Fig. S1). These results suggest that the abnormal expressions of calcium extrusion-related genes might be associated with COAD progression.

Next, we analyzed the relationship between the expressions of calcium extrusion-related genes and tumor stage. In COAD, the expression of SLC24A1 gradually decreased from stage I to IV, while SLC8A2 expression increased from stage I to IV in COAD, and the expressions of SLC8B1, SLC8A1, SLC8A3, SLC24A2, SLC24A3, SLC24A4 and SLC24A5 did not show significant differences among different tumor stage (Fig. 3). Taken together, these results indicated that calcium extrusion-related genes, especially SLC24A1 and SLC8A2, may be involved in the development of COAD.

Meanwhile, the relationship between expression of calcium extrusion-related genes and clinicopathological features in COAD was systematically analyzed. Results indicated that mRNA expressions of the calcium extrusion-related genes were not associated with age, gender, T stage, N stage and M stage in COAD (Figs. S2A–S2G). Then, the expression of calcium extrusion-related genes was evaluated using the Kaplan–Meier Plotter database to predict the prognosis of COAD patients. However, the mRNA expression of calcium extrusion-related gene had no prognostic values for COAD patients (Figs. S3A–S3D). Overall, these results suggest the individual calcium extrusion-related gene had not a good predictive value in COAD.

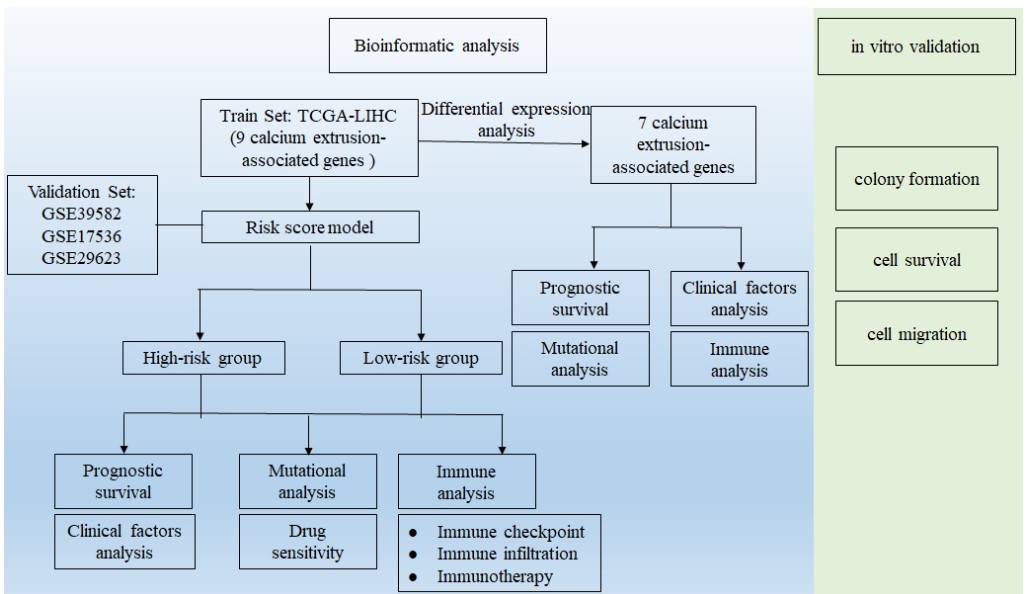

**Figure 1 Workflow diagram.** The flowchart graph of this study.

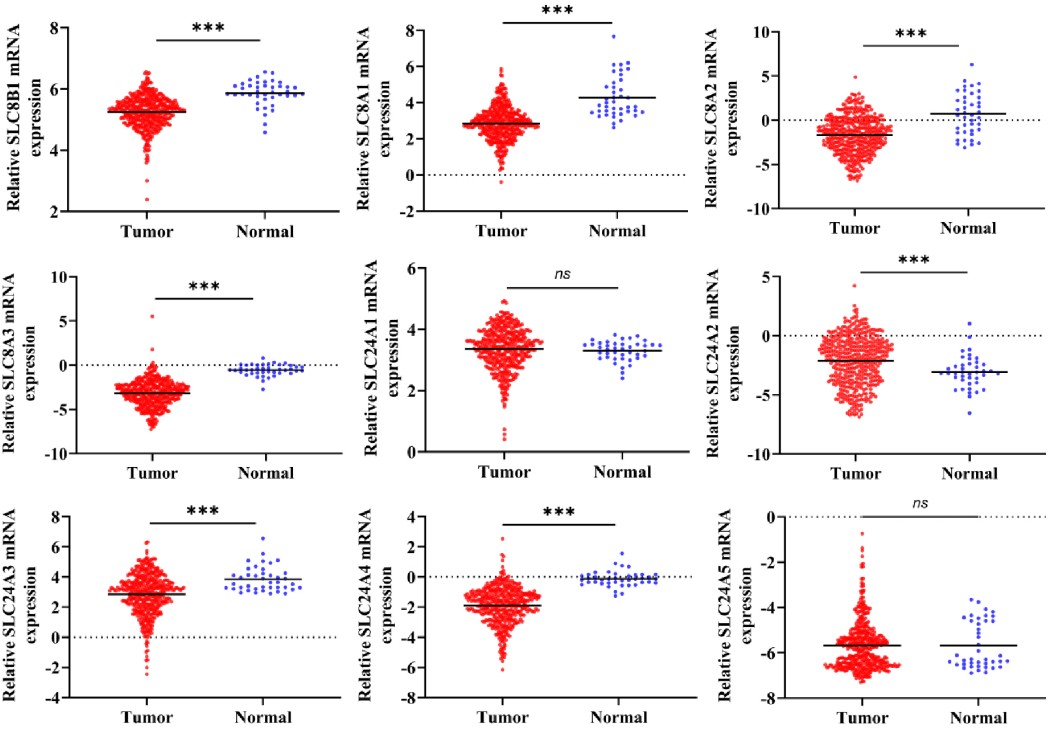

**Figure 2 The mRNA expressions of calcium extrusion-related genes in colorectal cancer tissues ($n = 430$) *versus* normal tissues ($n = 41$) from the datasets of TCGA.** *$P < 0.05$; **$P < 0.01$; ***$P < 0.001$.

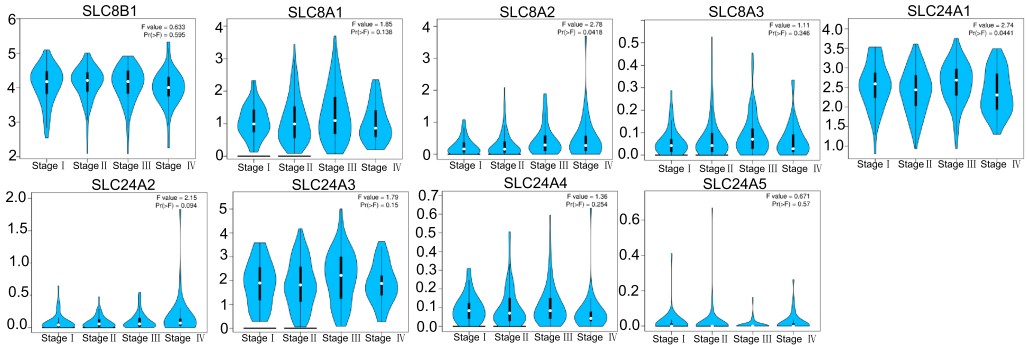

**Figure 3** Correlation between the mRNA expression of calcium extrusion-related genes and the pathological stage of patients.

## Gene mutation signature, protein-protein interaction network and correlation analysis of calcium extrusion-related genes in COAD

To further investigate the molecular characteristics of calcium extrusion-related genes, gene mutation, interaction network between proteins and correlation analysis were performed. In cBioPortal database, the calcium extrusion-related genes were rarely mutated; SLC8A1 showed the highest mutation rate of 6%, following by SLC24A2 and SLC24A3 with the same mutation rate of 4% (Fig. S4A). In addition, the expression of calcium extrusion-related genes had no significant correlation with TMB in COAD (Fig. S4B). We analyzed the correlation between the calcium extrusion-related genes with the protein-protein interaction network formed by SLC8B1, SLC8A1, SLC8A2 and SLC8A3. The expression of them was significantly correlated in COAD (Figs. S4C–S4D). The co-expression gene interactions opined the function of the calcium extrusion-related genes mainly focused on cellular calcium ion homeostasis, active transmembrane transporter activity and calcium ion transport (Fig. S4E).

## The association between mRNA expression of calcium extrusion-related genes and immune cell infiltration in COAD

We also studied the association between the expression of calcium extrusion-related genes and the infiltration of immune cells in COAD. CIBERSORT was used to determine the cellular composition of 22 immune cells, as shown in Fig. 4A and Fig. S5. Data indicated that the proportion of M0 macrophage cells were significantly different between the low-expression and high-expression groups divided by the median expression of SLC8A1, SLC8A2, SLC8A3, SLC24A2 and SLC24A3, respectively. There was a notable disparity in the percentage of M2 macrophage cells between the low and high categories of SLC8B1, SLC8A1, SLC24A2, SLC24A3, and SLC24A4. Additionally, there was a notable disparity in the proportion of resting CD4 memory T cells between the low-expression and high-expression categories of SLC8A2, SLC24A2, and SLC24A4. Furthermore, the distribution of activated CD4 memory T cells exhibited a significant contrast between the two groups of SLC8A1 and SLC24A3, respectively. Taken together, these results suggested

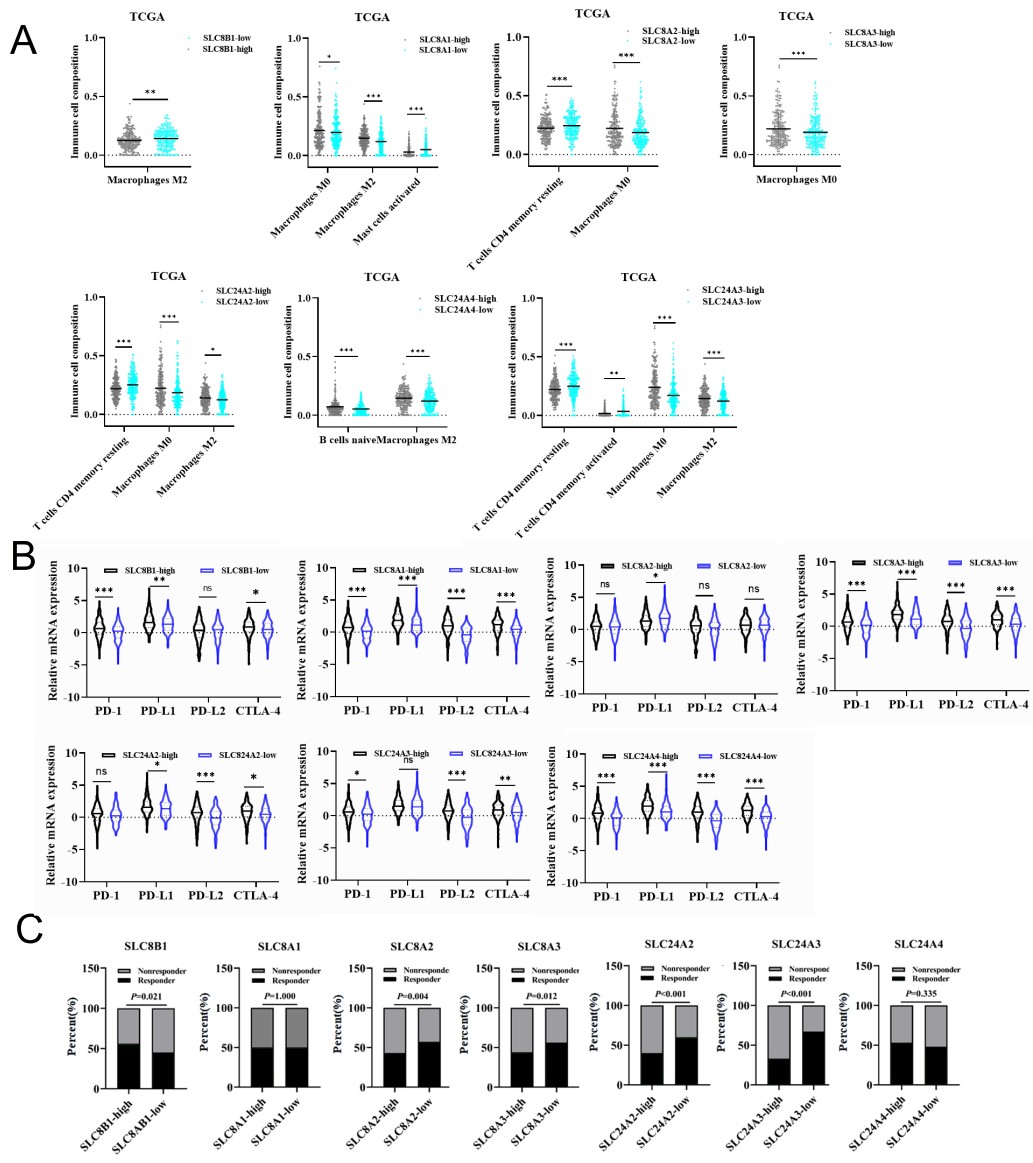

**Figure 4** **The relationships between the mRNA expression of calcium extrusion-related genes and immune cell infiltration, immune checkpoint and immunotherapy.** (A) The relationships between the mRNA expression of seven calcium extrusion-related genes and immune cell infiltration. (B) The expression of PD1, PD-L1, PD-L2 and CTLA4 in the low-expression and high-expression groups based on the median expression level of each calcium extrusion-related genes in COAD, respectively. (C) The distribution of responder and non-responder to immunotherapy in the low-expression and high-expression groups based on the median expression level of each calcium extrusion-related genes in COAD, respectively. $*P < 0.05$; $**P < 0.01$; $***P < 0.001$; ns stands for no significant change.

that the immune cell infiltration was closely related with the expressions of calcium extrusion-related genes, which needs to be clarified in the future.

## The association between the mRNA expression of calcium extrusion-related genes and immune checkpoint and immunotherapy response

Numerous studies reported that immune checkpoint molecules were closely associated with the response to immunotherapy in cancers. Therefore, we examined the expression levels of immune checkpoint molecules in different groups. Our results indicated that PD-1, PD-L1, PD-L2 and CTLA-4 expression level have significant differences between the low-and high-expression of almost all calcium extrusion-related genes, except for SLC8A2 (Fig. 4B). The potential of anti-PD1 and anti-CTLA4 immunotherapy response in the low-and high-expression groups of calcium extrusion-related genes was examined using TIDE. Our results showed that the SLC8B1-high expression group might respond better to immunotherapy, while the low expression group of SLC8A2, SLC8A3, SLC24A2 or SLC24A3 might respond better to immunotherapy (Fig. 4C). Additionally, the analysis of functional enrichment revealed that SLC8B1 and its co-expressed genes were involved in T cell activation, leukocyte intercellular adhesion, and the positive regulation of cell adhesion (Fig. S6). Altogether, these results demonstrated that the expressions of calcium extrusion-related genes might be associated with immunotherapy responses in COAD.

## Construction of calcium extrusion-related genes risk signature

Due to the failure prediction for prognosis of COAD using the individual calcium extrusion-related gene, we established a novel risk model based on expression level of calcium extrusion-related genes to predict prognostic outcomes of COAD patients. The TCGA-COAD dataset served as the training dataset, while three GEO datasets (GSE29623, GSE39582, and GSE17536) were utilized as the validation datasets. The results indicated that high-risk group had a worse prognosis for patients with COAD in training dataset (Figs. 5A and 5C) and validation datasets (Figs. 5B, 5D and Figs. S7A and S7B). Then, prognostic risk models were used in predicting 1-, 3-and 5-year overall survival, ROC curves were plotted with AUC values of 0.584, 0.623 and 0.636, respectively (Fig. 5E). Corresponding AUC values of 0.706, 0.675 and 0.704 were observed in the GSE17536 cohort; AUC values of 0.827, 0.637 and 0.792 were observed in the GSE29623 cohort, and AUC values of 0.564, 0.587 and 0.594 were observed in GSE39582 cohort (Fig. 5F and Fig. S7C). The clinical characteristics of the low-risk and high-risk groups were also analyzed and we found that age and T stage among the two risk groups were significantly different (Table 1). Altogether, these results indicated that the risk score model is effective in predicting the prognosis of COAD patients and patients with high risk have worse survival.

## Mutation profile of COAD patients in high- and low-risk groups

Next, we mapped the mutation landscapes of COAD in both the high-risk and low-risk groups. In the high-risk group, APC (70%), TP53 (56%), TIN (53%), KRAS (49%) and PIK3CA (35%) were the top five mutated genes, while APC (70%), TP53 (54%), TIN (52%), KRAS (37%) and MUC16 (34%) were the most frequently mutated genes in low-risk group (Fig. 6A). These gene mutations are shown in Fig. S8. Both groups shared APC, TP53, TTN, KRAS, PIK3CA, MUC16, SYNE1, FAT4, OBSCN, RYR2, ZFHX4, DNAH5, CSMD3,

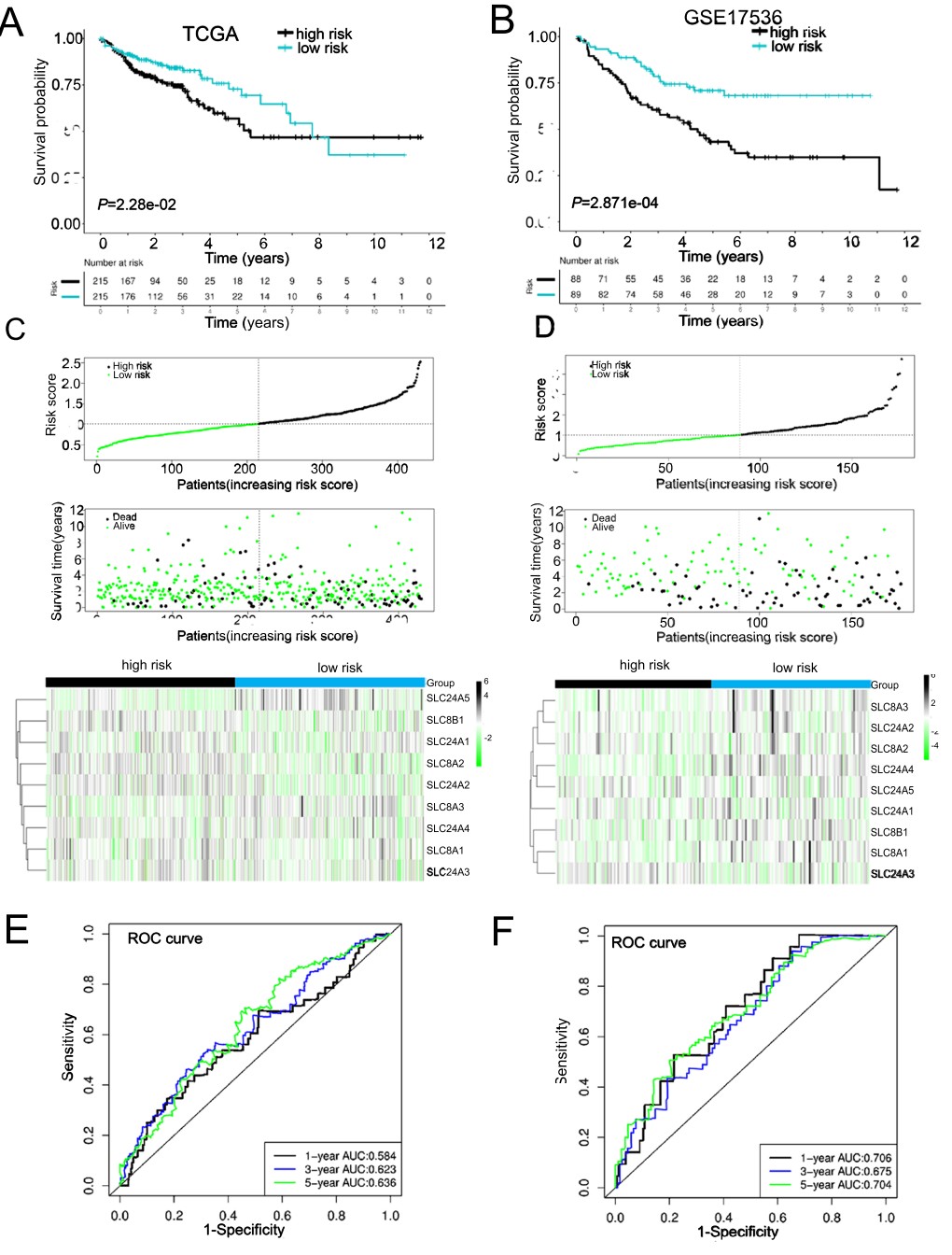

**Figure 5  Constructed and validated the calcium extrusion-related risk signature.** (A, B) K-M survival between the low-risk and high-risk group in the TCGA-STAD training cohort ($n = 430$) (A) and GSE17536 cohort ($n = 177$) (B). (C, D) Distribution of risk score, survival status (red dots indicate death, blue dots indicate living) and the gene expression of calcium extrusion-related genes in the TCGA-STAD training cohort (C), and GSE17536 cohort (D). (E, F) ROC curve and the areas under the curve (AUC) at 1, 3, and 5 years for the risk score in the TCGA-STAD training cohort (E) and GSE17536 cohort (F).

**Table 1 Clinical characteristics between low- and high-risk groups.**

| Variables | High risk<br>n = 215 | Low risk<br>n = 215 | P value |
|---|---|---|---|
| Age(years) | | | **0.033** |
| ≤66 | 107 | 85 | |
| >66 | 108 | 130 | |
| Gender | | | 0.333 |
| Female | 104 | 94 | |
| Male | 111 | 121 | |
| Tumor Stage | | | 0.955 |
| I–II | 119 | 119 | |
| III–IV | 91 | 90 | |
| N/A | 5 | 6 | |
| T | | | **0.028** |
| T1/2 | 34 | 52 | |
| T3/4 | 181 | 162 | |
| N/A | | 1 | |
| N | | | 0.624 |
| N0 | 124 | 129 | |
| N1/N2 | 91 | 86 | |
| M | | | 0.247 |
| M0 | 149 | 169 | |
| M1 | 33 | 27 | |
| MX | 33 | 19 | |
| Survival status | | | **0.036** |
| Alive | 159 | 177 | |
| Dead | 56 | 38 | |

**Notes.**
Bold indicates P value ≤ 0.05 was considered statistically significant.

FLG and DNAH11 mutations at high frequencies (Fig. 6B). Interestingly, the frequency of KRAS mutation were higher in patients of high-risk group, whereas MUC16 had a lower frequency of mutation in the high-risk group (Fig. 6C). The tumor mutation burden (TMB) was an important prognostic indicator for COAD patients. Therefore, we evaluated the relationship between risk score and TMB. The results demonstrated a significant negative correlation between risk score and TMB in COAD (Fig. 6D). The overall survival rate of high TMB patients was lower than that of low TMB patients (Fig. 6E). Moreover, in high TMB subgroup, high-risk group had worse prognosis than low-risk group (Fig. 6F). In conclusion, the above results indicated combination of risk score and TMB might be a more effective biomarker in predicting COAD prognosis.

## Calcium extrusion-related risk score was associated with immune signatures and immunotherapy responses in COAD

Using the CIBERSORT algorithm, we analyzed the proportion of 22 immune cells in the high-and low-risk groups and assessed relationship between risk score and tumor immune microenvironment (TIME). A significant increase in T cells regulatory (Tregs)

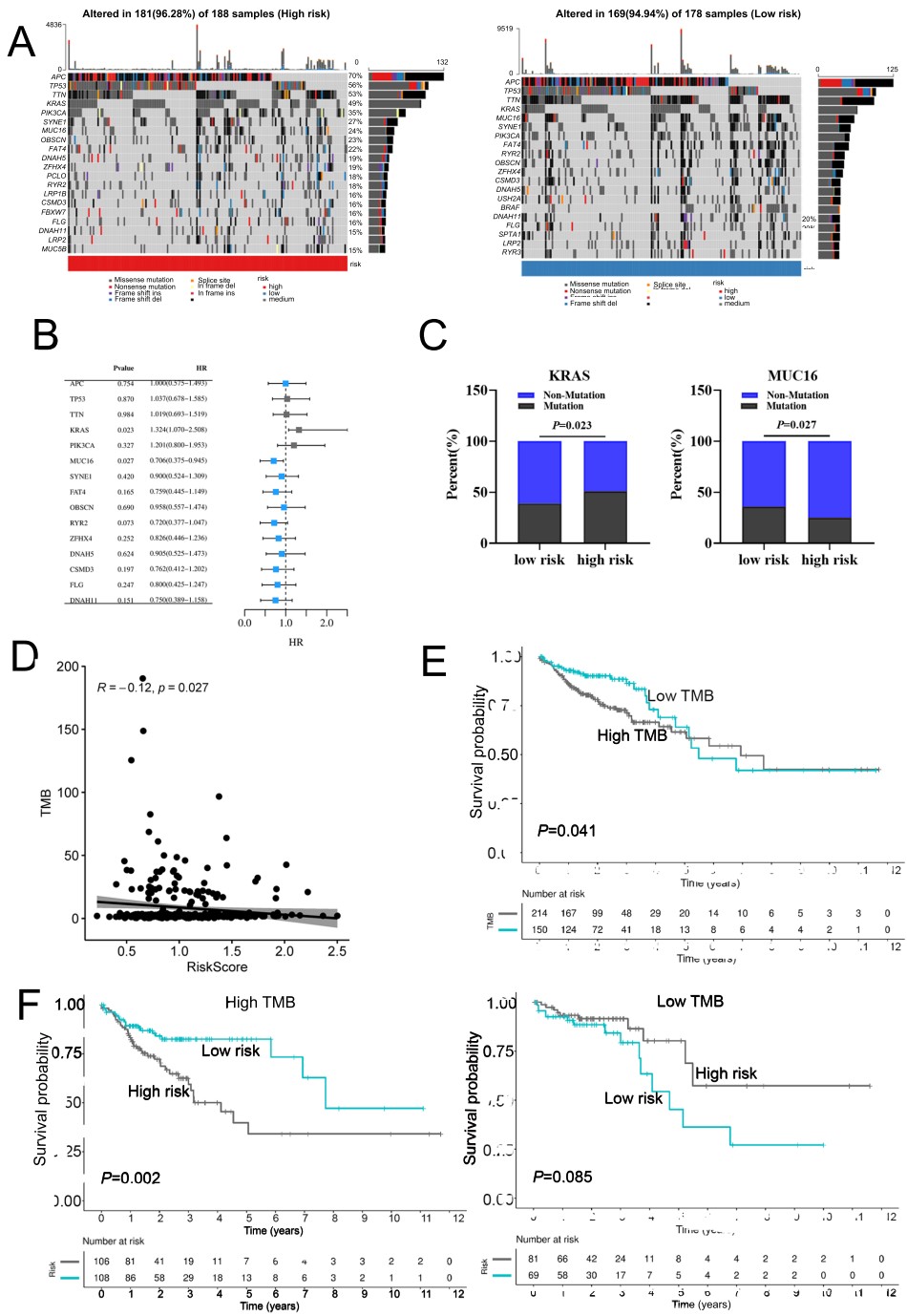

**Figure 6 The mutation profile in low-risk and high-risk groups.** (A) Mutation profile of COAD patients in low-risk and high-risk groups. (B) The top 15 genes with the higher mutation frequency variations of the mutation frequencies between the low-risk and high-risk groups. (C) The distribution of non-mutation and mutation samples of MUC16 and KRAS in the low-risk and high-risk group, respectively. (D)The relationship between the risk score and TMB. (E, F) Kaplan–Meier curves of the OS of patients in the high- and low-TMB groups in the training cohort.

and M0 macrophages was observed in the high-risk group, while the proportion of B cells naive, T cell CD4 memory resting, and M2 macrophages were remarkably decreased in the high-risk group (Fig. 7A, Fig. S9A). Similarly, risk score was positively correlated with majority signatures of Tregs (Fig. 7B). The relationship between the risk score and other immune cells signatures have been shown in Fig. S9B. Consisted with previous studies, we found that high-risk groups had lower immune score than low-risk group (Fig. 7C). Collectively, these findings suggest that the increased infiltration of Tregs-involved tumor immunosuppressive microenvironment might contribute to the poor survival for patients with COAD.

In addition, our data showed that six inhibitory immune checkpoints and eight stimulatory immune checkpoints were altered in high-risk group. The expressions of inhibitor immune checkpoints including CD160, CD244, IL10, KIR2DL1 and KIR2DL3, were declined in high-risk groups. The expressions of stimulatory immune checkpoints including ICOSLG, PVR, TNFSF13 and TNFSF15 were increased in high-risk groups (Fig. 7D). In patients with COAD, the effectiveness of risk scoring in predicting the outcomes of immunotherapy was evaluated by the TIDE algorithm. Our findings indicated that the high-risk group exhibited a lower response rate to immunotherapy in comparison to the low-risk group, but this disparity did not achieve statistical significance (Fig. 7E). Subsequently, the PRJEB25780 cohort, comprising of 45 individuals diagnosed with advanced stomach cancer receiving PD-L1 inhibitor therapy, was employed to determine the effectiveness of the risk signature associated with calcium extrusion in accurately predicting the outcomes of immunotherapy. The low-risk group showed a considerably higher immunotherapy response rate compared to the high-risk group (Fig. 7F). As shown in Fig. 7G, the immunotherapy response rate was significantly higher in the subgroup with a high-immune score than those with a low-immune score. Notably, within the high-immune score subgroup, the low-risk group demonstrated a higher immunotherapy response rate (63.00%) compared to the high-risk group (58.00%) (Fig. 7H). These findings indicated that risk score alone or combined with risk score and immune score could predict the immunotherapy responses in patients with COAD.

TMB has been shown to serve as a valuable biomarker for the purpose of selecting immune checkpoint blockade in various cancer types. The subgroup with a high TMB exhibited a slightly higher immunotherapy response rate (51.00%) compared to the subgroup with a low TMB (44.00%), although this disparity did not reach statistical significance (Fig. S9D). Moreover, our findings indicated that the rate of immunotherapy response in the low-risk group (56.00%) was comparatively higher than that observed in the high-risk group (46.00%) within the high tumor mutational burden (TMB) subgroup, despite the lack of statistical significance (Fig. S9E). The phenotype linked to microsatellite instability-high (MSI-H) is a unique category of tumors that display an increased vulnerability to immunotherapy. Notably, A significantly higher immunotherapy response rate was observed in the MSI-H subgroup compared to the MSS subgroup (Fig. S9F). Moreover, our findings indicated that the immunotherapy response rate in the low-risk group (68.00%) was comparatively higher than that observed in the high-risk group (61.00%) within the MSI-H subgroup, although this difference did not reach

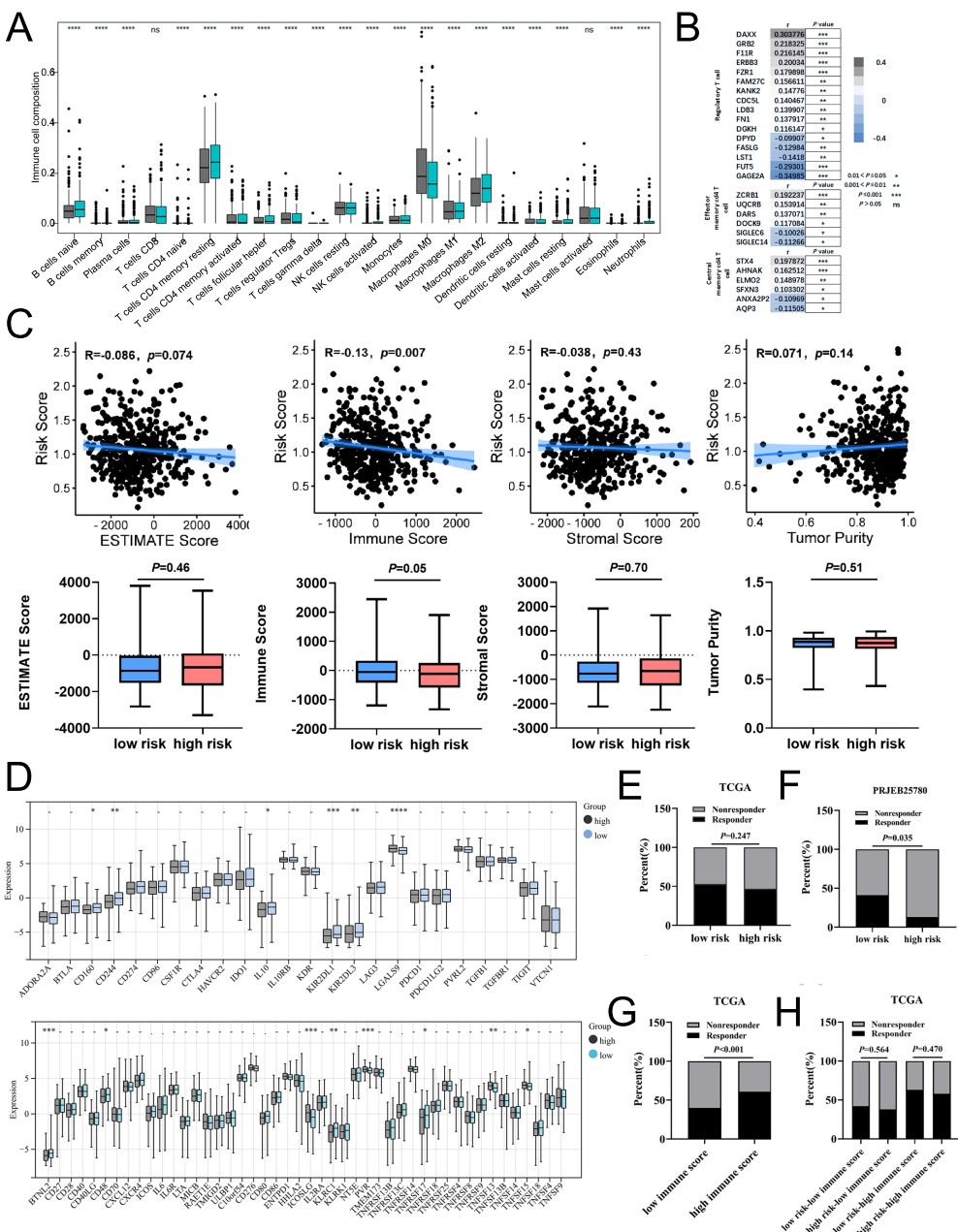

**Figure 7  The associations of risk scores of immune signatures and immunotherapy responses.** (A) Comparison of the difference in immune cell fraction between the low-risk and high-risk group. (B) The relationship between risk score and the expression of immune cell marker genes. (C) The correlation between ESTIMATE scores, stromal, immune, and tumor purity and risk score in training set. (D) The relationship between the risk score and expression of immune checkpoints. (E) TIDE predicted the proportion of patients with response to immunotherapy in low-risk and high-risk groups. (F) The proportion of patients with response to immunotherapy in low-risk and high-risk groups in the PRJEB25780 immunotherapy cohort (45 patients with advanced gastric cancer who had received PD-L1 inhibitor treatment). (G) TIDE predicted the proportion of patients with response to immunotherapy in low-immune

**Figure 7 (…continued)**
score and high-immune score groups. (H) TIDE predicted the proportion of patients of four groups based on the risk score and immune score with response to immunotherapy. *P < 0.05; **P < 0.01; ***P < 0.001; ****P < 0.0001; ns stands for no significant change.

**Table 2   Drug information of the top 10 sensitivity in low-risk group.**

| Drug name | Synonyms | Targets | Target pathway |
|---|---|---|---|
| BMS-754807 | BMS754807 | IGF1R, IR | RTK signaling |
| Dabrafenib | GSK2118436, Tafinlar | MEK1, MEK2 | ERK MAPK signaling |
| Daporinad | APO866, FK866 | NAMPT | Metabolism |
| PLX-4720 | PLX4720 | BRAF | ERK MAPK signaling |
| SB216763 | SB-216763 | GSK3A, GSK3B | WNT signaling |
| UMI-77 | UMI 77 | MCL1 | Apoptosis regulation |
| MIM1 | MIM-1 | MCL1 | Apoptosis regulation |
| LJI308 | NA | RSK2, RSK1, RSK3 | PI3K/MTOR signaling |
| PD0325901 | PD-0325901 | MEK1, MEK2 | ERK MAPK signaling |
| Telomerase Inhibitor IX | MST-312 | Telomerase | Genome integrity |

statistical significance (Fig. S9G). Taken together, these findings suggested that combined with risk score, immune score, TMB, or microsatellite status might be a potential strategy to predict immunotherapy response in patients with COAD.

## Risk score predicts therapeutic benefits in COAD

In order to assess the efficacy of risk score as a biomarker for prognosticating drug response in COAD, the IC50 value of 198 drugs was deduced for TCGA-COAD patients. The results showed that the IC50 values of BMS-754807, Dabrafenib, Daporinad, PLX-4720, SB216763, UMI-77, MIM1, LJI308, PD0325901, and Telomerase inhibitor IX in the low-risk group were significantly lower than the high-risk group (Fig. 8A and Fig. S10). These results demonstrated that patients in the low-risk group were relatively sensitive to these agents. However, patients in high-risk group exhibited a higher sensitivity to AZD6482, KU-55933, PF-4708671, Acetalax, GSK2699962A, RO-3306, NU7441, Ribociclib, Sapitinib, and Taselisib (Fig. 8B and Fig. S10). In conclusion, the results indicated that patients in different group might be sensitive to various drugs, which was of great importance for the development of individualized treatment and clinical medication guidance. Tables 2 and 3 provided comprehensive details about the top 10 vulnerable medications for both the low-risk and high-risk subgroup.

## Function analysis of the previously unreported model genes SLC8A3, SLC24A2, SLC24A3 and SLC24A4 in CRC

Previous studies have shown that SLC8A1, SLC8A2, and SLC8B1 contribute to the initiation and progression of cancer (Muñoz et al., 2015; Qu et al., 2017; Pathak et al., 2020). Therefore, we examined previously unreported model genes for their functional roles (SLC8A3, SLC24A2, SLC24A3 and SLC24A4) in CRC cells. We first investigated the mRNA expression level of SLC8A3, SLC24A2, SLC24A3 and SLC24A4 in different

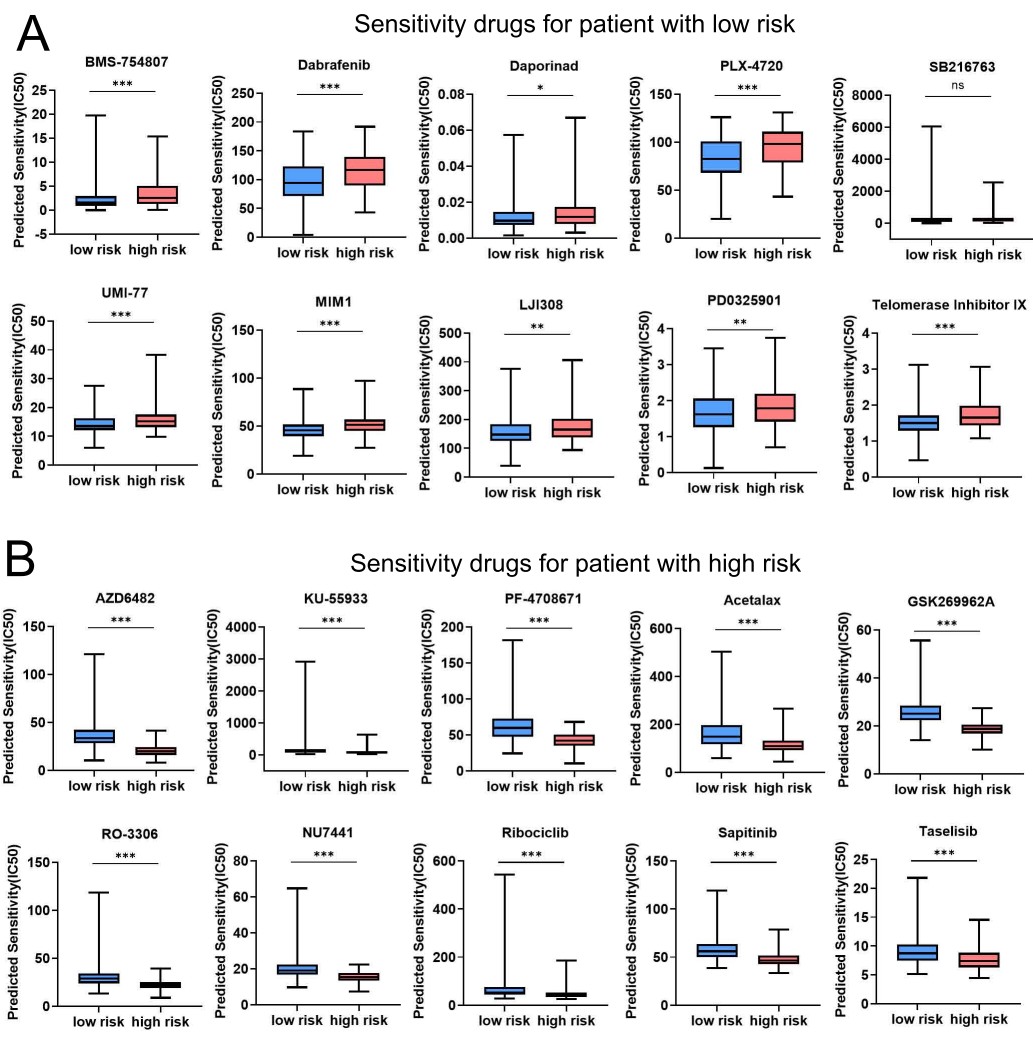

**Figure 8   Risk score predicts drug therapeutic benefits in COAD.** (A) The top 10 sensitivity drugs for patients with low-risk. (B) The top 10 sensitivity drugs for patients with high-risk. *$P < 0.05$; **$P < 0.01$; ***$P < 0.001$.

types of CRC cells. Then, we assessed cell viability and colony formation in those overexpression or knockdown RKO cell, which exhibited a relative median expression level of SLC8A3, SLC24A2, SLC24A3 and SLC24A4 (Fig. S11A). To validate the efficiency of cell transfection, western blots were conducted, and the results were presented in Fig. 9A and Fig. S11B. The results indicated that the downregulation of SLC8A3 or SLC24A4 expression promoted RKO cells growth *in vitro* (Figs. 9B–9C), while the overexpression of SLC8A3 or SLC24A4 inhibited RKO cells growth *in vitro* (Figs. S11C–S11D). The knockdown or overexpression of SLC24A2 and SLC24A3 unexpectedly exhibited a weak effect on CRC growth (Figs. 9B–9C and Figs. S11C–S11D). Moreover, we assessed CRC cell migration in those overexpression or knockdown RKO cell by scratch wound healing assay. The results revealed that the migration ability of RKO cells was markedly increased when

**Table 3  Drug information of the top 10 sensitivity in high-risk group.**

| Drug name | Synonyms | Targets | Target pathway |
| --- | --- | --- | --- |
| AZD6482 | AK-55409 | PI3Kbeta | PI3K/MTOR signaling |
| KU-55933 | NA | ATM | Genome integrity |
| PF-4708671 | PF4708671 | S6K1 | PI3K/MTOR signaling |
| Acetalax | Oxyphenisatin acetate | NA | NA |
| GSK269962A | NA | ROCK1, ROCK2 | Cytoskeleton |
| RO-3306 | NA | CDK1 | Cell cycle |
| NU7441 | KU-57788, NU-7432 | DNAPK | Genome integrity |
| Ribociclib | LEE011, NVP-LEE011, LEE011-BBA | CDK4, CDK6 | Cell cycle |
| Sapitinib | AZD8931 | EGFR, ERBB2, ERBB3 | EGFR signaling |
| AZD8055 | GDC-0032, GDC0032, RG7604 | PI3K (beta sparing) | PI3K/MTOR signaling |

SLC8A3 or SLC24A4 was downregulated. By contrast, there was a significant decrease in the migration capability of CRC cells in SLC8A3 or SLC24A4 overexpression cells (Fig. 9D and Fig. S11E). However, cell migration was not affected by the knockdown or overexpression of SLC24A2 and SLC24A3 (Fig. 9D and Fig. S11E). The results showed that SLC24A2 and SLC24A3 did not affect CRC cell proliferation and migration. Altogether, these results suggest that SLC8A3 and SLC24A4 may be involved in CRC growth and metastasis as tumor suppressor genes.

## DISCUSSION

Many studies have revealed the important functions of calcium signaling in tumor progression (*Bettaieb et al., 2021*; *So et al., 2019*; *Silvestri et al., 2023*; *Yang et al., 2019*; *Monteith, Prevarskaya & Roberts-Thomson, 2017*; *Iamshanova, Pla & Prevarskaya, 2017*; *Marchi & Pinton, 2016*). NCLX, $Na^+/Ca^{2+}$ exchangers (NCXs) and $Na^+/Ca^{2+}$–K+ exchanger(NCKXs), belonging to the superfamily of $Ca^{2+}$/Cation antiporter, are crucial components in $Ca^{2+}$ efflux mechanism. They are essential for cell function and survival by maintaining $Ca^{2+}$ homeostasis (*Khananshvili, 2013*; *Rodrigues, Estevez & Tersariol, 2019*; *Schnetkamp, 2013*; *Luongo et al., 2017*). To date, the prognostic value of calcium extrusion-related genes in colon cancer have not been studied systematically. Thus, it is necessary to examine the potential capacity of calcium extrusion-associated genes in developing a prognostic risk model in COAD.

We investigated the mRNA level of nine calcium extrusion-related genes in COAD. Intriguingly, most of their mRNA expression level were downregulated in colon cancer tissues, compared with normal colon tissues. We then analyzed the prognostic value of individual calcium extrusion-related genes in COAD. Unfortunately, the mRNA expression of individual genes failed to predict prognosis in COAD patients. Thus, we built up a risk signature relying on the calcium extrusion-related genes. Based on median risk scores, patients were split into high-risk and low-risk groups. K-M survival analysis shown that patients with a low risk lived longer than those with a high risk. Moreover, there was a relatively high AUC value for the risk model. These results indicated that the risk profile contributed to a better risk stratification and prognostic prediction of patients with COAD

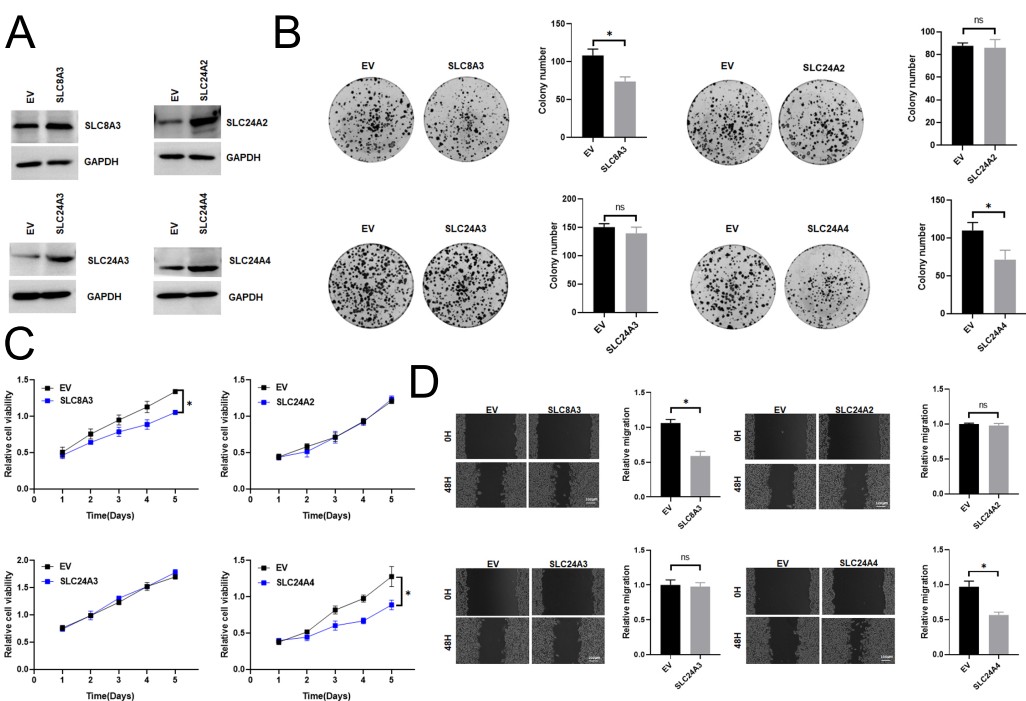

**Figure 9** **The functional role of SLC8A3, SLC24A2, SLC24A3 and SLC24A4 in CRC cells.** (A) Western blotting analysis to measure SLC8A3, SLC24A2, SLC24A3 and SLC24A4 protein expression in RKO cells, treated as indicated. (B) CCK8 assays (C) Scratch wound healing (D) Colony formation assay of RKO cells. Treatment as indicated. *$P < 0.05$; ns stands for no significant change.

(Fig. 10). The mitochondrial Na$^+$/Ca$^{2+}$ exchanger (NCLX) encoded by SLC8B1 is believed to be the primary mechanism for mtCa$^{2+}$ (*Luongo et al., 2017*). The loss of NCLX causes mtCa$^{2+}$ overload, resulting in accumulation of mitochondrial ROS, which activate HIF1 $\alpha$ signaling and promote the metastasis of tumor cells (*Pathak et al., 2020*). *Tangeda et al. (2022)* reported that mitochondrial Lon upregulation contributes to cisplatin resistance in cancer cells. The mitochondrial Lon interacts with and activates NCLX activity to promote calcium release into cytosol, which increase Bcl-2 and IL-6 expression through calcium-dependent PYK2-SRC-STAT3-IL-6 pathway, leading to resistance to cisplatin. Interestingly, we found that SLC8B1 mRNA level was significantly reduced in colon cancer.

The SLC8 gene family contains three genes, SLC8A1, SLC8A2, and SLC8A3, which were involved in Ca$^{2+}$ efflux across cell membranes (*Khananshvili, 2013*). In exchange for sodium ions, calcium ions are transported out of the cell by Na$^+$/Ca$^{2+}$ exchangers (NCX) (*Chovancova et al., 2020*). *Liu et al. (2022)* found the NCX1 expression of melanoma cell is upregulated compared with normal melanocytes, and pharmacological inhibition of the NCX1 leads to Ca$^{2+}$-related cell death could suppressing the growth of melanoma cells. However, our results showed that the SLC8A1 expression level is decreased in colon cancer. We speculated that such differences might result from different types of cancer and samples. *Qu et al. (2017)* reports the SLC8A2 gene acts as a tumor suppressor to inhibit invasion, angiogenesis and the growth of glioblastoma through ERK1/2-NF$\kappa$B-MMPs/uPA-uPAR

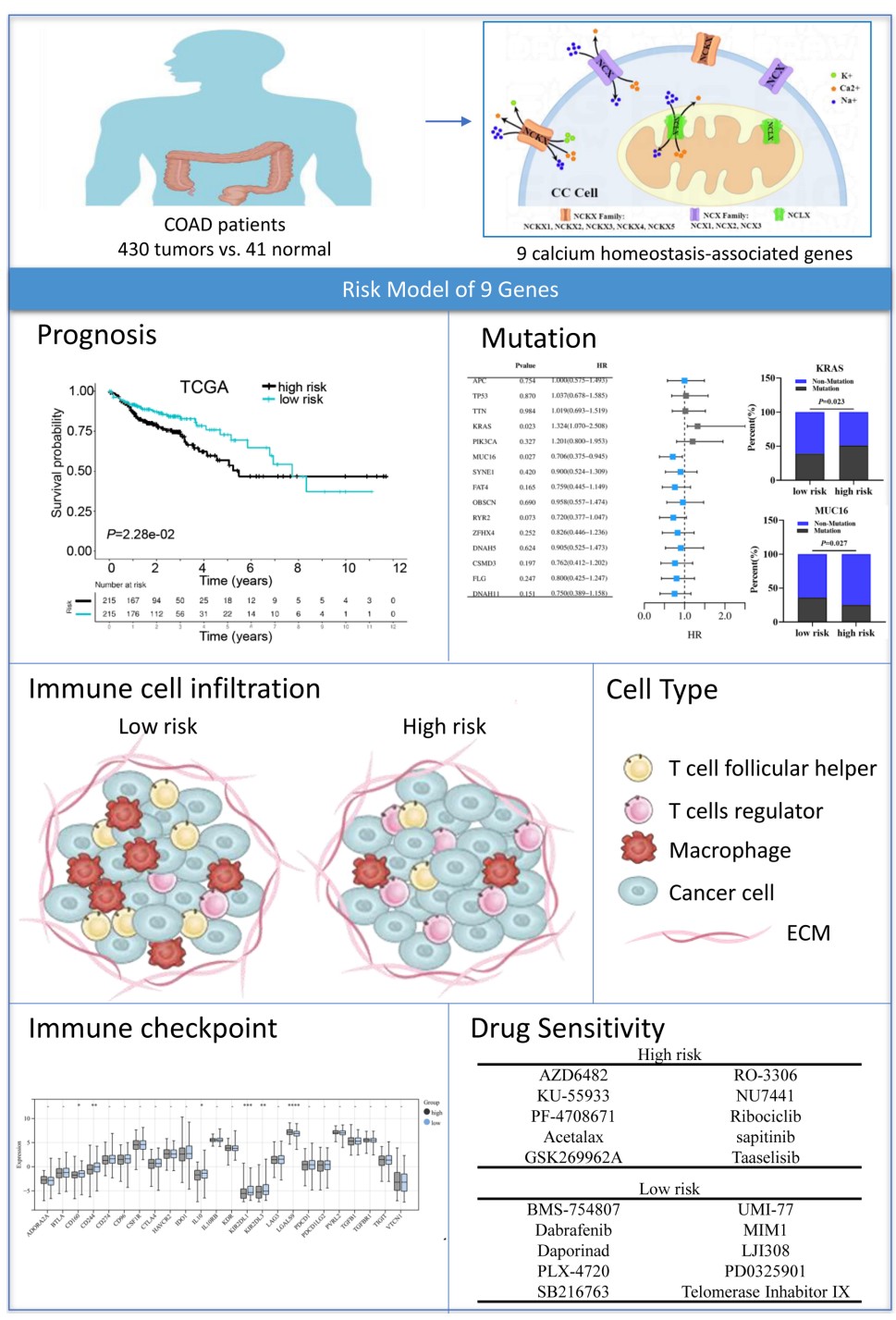

**Figure 10  Graph summarization.** The work summary graph of this study. The diagrammatic drawing of COAD patients (ID:SRRAU1cab1) , calcium extrusion-related molecules in colon cancer cell (ID:TTWPY18b8f), and immune cell infiltration diagram (ID:APWlOac75a) were drawn using Figdraw software (https://www.figdraw.com/static/index.html#/paint_index). The representational pictures of prognosis, mutation, immune checkpoint and drug sensitivity were from Figs. 5A, 6B–6C, 7D and 8A–8B, respectively, in the present study.

signals. Consistent with our results, SLC8A2 expression level was decreased in colon cancer. Human SLC24 genes include NCKX1-5. NCKXs are involved in a variety of biological functions (*Schnetkamp, 2013*; *Jalloul et al., 2020*). *Tran et al. (2021)* reported that the NCKX3 KO mice promoted DSS-induced colitis *via* activation of p53 and NFkB signaling pathways, suggesting that NCKX3 plays a critical role in the immune response. Notably, our results exhibit that the expressions of SLC24A3 in COAD was significantly lower compared to normal colon tissues.

Recently, it has been suggested that NCLX B-cell specific knockout mice showed reduced germinal center B cell responses following foreign antigen and pathogen driven immune responses (*Emrich et al., 2022*). The inhibition of NCLX reversed the salt-induced Treg dysfunction (*Côrte-Real et al., 2023*). In addition, we found a significant increase in the proportion of T cells regulatory (Tregs) and M0 macrophages in the high-risk group, while the proportion of B cells naive, T cell CD4 memory resting, and M2 macrophages decreased, suggesting that calcium extrusion-related genes could be a key role in immune cell infiltration. Then, we used the TIDE algorithm to predict the potential efficacy of anti-PD1 and anti-CTLA4 immunotherapy in the high-risk and low-risk groups of COAD patients. There was a higher number of immunotherapy responders in the low-risk group than in the high-risk group. Overall, our results suggest that calcium extrusion-related genes might play an essential role in the immune response, and patients with low risk might exhibit a more favorable response to PD-1 and CTLA-4 treatment.

The association of SLC8A3, SLC24A2, SLC24A3 and SLC24A4 with cancer is still unknown. Therefore, we tested the function roles of the previously unreported model gene in CRC cells. The results showed that the knockdown of SLC8A3 and SLC24A4 could promote the growth and migration of RKO cells *in vitro*, while overexpression of SLC8A3 and SLC24A4 inhibited the growth and migration of RKO cells *in vitro*. However, SLC24A2 and SLC24A3 have little effect on RKO cells. Our data indicate that the newly identified tumor suppressor gene SLC8A3 and SLC24A4 has potential for clinical application and may offer therapeutic targets for patients with CRC.

This study is subject to certain limitations. Firstly, a prognostic model based on the expression of calcium extrusion-related genes in COAD was developed using bioinformatics analysis; however, validation in clinical specimens was not possible due to the lack of such specimens and relevant information. Future research should focus on collecting patient samples and relevant clinical data to validate the efficiency of the model. Secondly, the key signaling pathways regulated by SLC8A3 and SLC24A4 that involved in CRC growth and metastasis were not deeply investigated. While these genes have been confirmed to have a significant association with the growth and migration of colon cancer in cellular function assays, further research is required to investigate the potential regulatory mechanisms. It is imperative and rational to explore the function and mechanisms of these potential signature genes (SLC8A3 and SLC24A4) in CRC progression using experimental animal models and tumor cells derived from patients in future studies.

In conclusion, we constructed a prognostic risk score model based on calcium extrusion-related genes in COAD. Functionally, the risk score was highly correlated to the immune cell infiltration and level of immune checkpoint molecules of COAD patients. Combined

analysis for risk score and immune score, or TMB, or MSS/MSI could predict the response to immunotherapy in COAD. In addition, we validated the function of four unreported genes by several experiments, and our results suggested that SLC8A3 and SLC24A4 could play a potentially critical role in the pathogenesis and progression of CRC. Our results may provide a basis for further study on their mechanisms in calcium extrusion and colon cancer progression.

### Funding
This work was supported by the National Natural Science Foundation of China (No. 81902513), and the Natural Science Foundation of Shanghai Municipality (No. 22XD1422300) and Shanghai Municipal Health Commission (No. 2022XD053). The funders had no role in study design, data collection and analysis, decision to publish, or preparation of the manuscript.

### Grant Disclosures
The following grant information was disclosed by the authors:
National Natural Science Foundation of China: 81902513.
Natural Science Foundation of Shanghai Municipality: 22XD1422300.
Shanghai Municipal Health Commission: 2022XD053.

### Competing Interests
The authors declare there are no competing interests.

### Author Contributions
- Mingpeng Jin performed the experiments, analyzed the data, prepared figures and/or tables, and approved the final draft.
- Chun Yin performed the experiments, analyzed the data, prepared figures and/or tables, and approved the final draft.
- Jie Yang performed the experiments, authored or reviewed drafts of the article, and approved the final draft.
- Xiaoning Yang analyzed the data, authored or reviewed drafts of the article, and approved the final draft.
- Jing Wang performed the experiments, prepared figures and/or tables, and approved the final draft.
- Jianjun Zhu conceived and designed the experiments, analyzed the data, authored or reviewed drafts of the article, and approved the final draft.
- Jian Yuan conceived and designed the experiments, performed the experiments, authored or reviewed drafts of the article, and approved the final draft.

### Data Availability
   The data is available at NCBI GEO: GSE17536, GSE29623, and GSE39582.
   The raw data are available in the Supplemental Files.

## Supplemental Information

Supplemental information for this article can be found online at http://dx.doi.org/10.7717/peerj.17582#supplemental-information.

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
