# Peer review of "Identification and validation of calcium extrusion-related genes prognostic signature in colon adenocarcinoma"

_PeerJ, doi:10.7717/peerj.17582_

## Round 0.1 · original submission · Major Revisions

The reviewers found your manuscript interesting, however they had a number of significant concerns that need to be addressed.

First, the reviewers suggested that your abstract be rewritten to more clearly and succinctly state the aims, results and conclusions of your study. The reviewers requested that more detail be provided in the materials and methods section, specifically with regard to the clinical and pathological information of the TCGA data you utilized (such as inclusion/exclusion criteria, the number of patients in the disease and control groups), as well as more information on the cell lines used for your analyses.

The reviewers also commented that your introduction needs to be rewritten to more clearly state why you performed the study and to more succinctly describe and discuss the genes you studied. The results section needs to have a more clear and concise representation of your main findings. Additionally, the reviewers suggested that p-values be included in all figures and that how the fact that a number of the p-values shown (e.g. in Figure 3) fall below the threshold of statistical significance impacts the validity of your findings needs to be addressed.

Regarding the discussion section, the reviewers suggested including mechanistic insights into how the genes you identified contribute to the progression of COAD as well as a more in-depth discussion of the pathways involved. Your discussion also needs to include a more detailed description of the limitations of your study as well as how these limitations affect your results.

Please, submit a detailed rebuttal that shows where and how you have taken all comments and suggestions into consideration. If you do not agree with some of the reviewers’ comments or suggestions, please explain why. Your rebuttal will be critical in making a final decision on your manuscript. Please, note also that your revised version may enter a new round of review by the same or by different reviewers. Therefore, I cannot guarantee that your manuscript will eventually be accepted.

**Language Note:** The review process has identified that the English language must be improved. PeerJ can provide language editing services - please contact us at [email protected] for pricing (be sure to provide your manuscript number and title). Alternatively, you should make your own arrangements to improve the language quality and provide details in your response letter. – PeerJ Staff

Reviewer 1 ·

Basic reporting

The manuscript demonstrates clear and professional English usage, maintaining a consistent and understandable narrative throughout. References to relevant literature are adequately incorporated, providing sufficient background in the field. The structure of the article, including figures, tables, and the sharing of raw data, adheres to professional standards, contributing to the overall clarity and comprehensiveness of the work. However, while the manuscript is self-contained and presents relevant results aligned with its hypotheses, there is room for improvement in demonstrating more compelling and novel insights, particularly in the interpretation and discussion of these results. This would enhance the manuscript's contribution to the field and its overall impact.

Experimental design

The manuscript aligns well with the journal's Aims and Scope, showcasing original primary research. It contributes valuably to the field and displays a high level of academic rigor and relevance.The research question, while defined, lacks depth and significance.The investigation appears to fall short in terms of technical and ethical standards. The lack of rigor in the research approach raises concerns about the validity and reliability of the findings.The methods section of the manuscript is not adequately detailed.

Validity of the findings

The manuscript falls short in assessing its impact and novelty. It lacks the necessary innovation, and the research does not sufficiently support its conclusions. Some of the data presented in the article fail to robustly support the conclusions drawn. The conclusions drawn in the manuscript are not well-supported by the research conducted.

Additional comments

Firstly, your research does not seem to demonstrate a unique innovation or a deep understanding. In the process of constructing a prognostic model, it is common to first collect a large number of relevant genes, then select those with prognostic significance, and finally analyze them through lasso-cox regression. However, this critical step appears to have been overlooked in your study. Furthermore, most of the data and figures provided, such as Figure 3, do not exhibit the necessary statistical significance (such as p-values), which calls into question the reliability of your results.

Secondly, the validation of your model in a large sample did not perform as well as expected. Generally, larger samples should provide more robust results. This may indicate certain limitations in your model or methodological issues, as seen in Figure 5.

Finally, while your research incorporates various analytical methods, it overall lacks original thinking and deep theoretical support, as shown in Figures 6-8. A successful study should not only be a compilation of techniques but more importantly, should demonstrate the author's unique insights and profound understanding of the subject.

Reviewer 2 ·

Basic reporting

The abstract needs to be improved completely. Organise the abstract so that you briefly introduce the aims of the work, the results and the conclusions. Keep it short but to the point. Also, if something is related to cell lines, it should be clearly stated and it should be specified which cell lines are involved.
Explain why you did this research in the introduction. Also introduce the reader to the topic. The introduction is very poor and needs to be corrected.

Experimental design

In the materials and methods, please describe the cell lines used and why you chose them.

Please provide a brief diagram of your results so that you can make sense of them.

Please also send original photographs of the blots - without using computer programs.

Validity of the findings

What conclusions have you drawn from your research? Is there any chance of translating this into a clinical context? Please refer to the relevant literature.

Reviewer 3 ·

Basic reporting

This study aimed to develop a prognostic risk model using calcium extrusion-related genes in colon adenocarcinoma (COAD). The authors developed a risk model combining multiple calcium extrusion genes that stratifies COAD patients into high and low risk groups with significant survival differences. The risk model was validated in multiple external datasets. Additionally, the authors provide insight into the interaction between calcium signaling and immune phenotypes in colon cancer.

Overall the study design and findings appear novel and meaningful. The authors should address important and minor issues to strengthen the quality before the paper is considered for publication.

1. The manuscript requires improvement in its writing. The authors need to comprehensively enhance the clarity and readability of the entire manuscript. Some notable issues as examples:
a) The abstract is too long and could be made more concise – some result details could be trimmed or consolidated to sharpen the focus.
b) In the methods, more specifics could be provided on the patient clinical/pathological data extracted from TCGA.
c) In the introduction and results, the author should briefly introduce the genes they studied: SLC8B1, SLC8A1, SLC8A2, SLC8A3, SLC24A3, and so on.
d) While the discussion emphasizes the correlation between altered expression of calcium extrusion-related genes and clinical outcomes, it falls short in providing mechanistic insights into how these genes contribute to COAD progression. A more detailed exploration of the molecular pathways and interactions involved would enhance the discussion.
e) While the limitations are briefly mentioned at the end of the discussion, a more detailed acknowledgment of study limitations and their potential impact on the interpretation of results would enhance transparency and reliability.

2. The study involves complex bioinformatic analyses, and the presentation of results could be challenging for non-experts. A clearer and more concise presentation of key findings would enhance the accessibility of the study.

3. I didn’t find the legends of supplement figures.

4. The gene mutation analysis reveals low mutation rates for calcium extrusion-related genes. However, the study does not delve into the functional consequences of these mutations or explore potential associations with tumorigenesis.

Experimental design

The data sourced from the TCGA and GEO-COAD databases requires additional information regarding the inclusion and exclusion criteria. The number of patients (tumor) and normal people are unclear, the author should clarify it in results or figure legends.

Validity of the findings

The study examines mRNA expression patterns, mutations, and their correlation with COAD risk, prognosis, immune cell infiltration, immune checkpoint responses, and immunotherapy outcomes. Nevertheless, there is a need for more thorough validation of these findings. Furthermore, the study should explicitly address potential biases and limitations related to the construction of the risk model.

---

## Round 0.2 · accepted · Accept

Thank you for thoroughly addressing the reviewers' comments and thus greatly improving your manuscript.

Reviewer 1 ·

Basic reporting

No comments.

Experimental design

No comments.

Validity of the findings

While I appreciate the detailed explanations and revisions made to the paper, I regret to inform you that I still have concerns regarding the study's design. This leads me to believe that the research lacks sufficient novelty and some of the results do not seem entirely reliable. Additionally, the cell study appears too basic to convincingly support the conclusions.

Additional comments

No comments.

Reviewer 2 ·

Basic reporting

the author has made efforts to change his manuscript according to my advice

Experimental design

the author has made efforts to change his manuscript according to my advice

Validity of the findings

the author has made efforts to change his manuscript according to my advice

Additional comments

the author has made efforts to change his manuscript according to my advice

Reviewer 3 ·

Basic reporting

The author answered all my questions and concerns. The manuscript is ready for publication.

Experimental design

No comment

Validity of the findings

No comment